# Effect of Spray Parameters on Electrical Characteristics of Printed Layer by Morphological Study

**Gye Hyeon Kim [1,2], Eun Ae Shin [2], Je Young Jung [2], Jun Young Lee [1,\*] and Chang Kee Lee [2,\*]**

[1] School of Chemical Engineering, Sungkyunkwan University, Seoul 16419, Korea; khsherry@g.skku.edu
[2] Korea Packaging Center, Korea Institute of Industrial Technology, Cheonan 15588, Korea; sea@kitech.re.kr (E.A.S.); jjy5782@kitech.re.kr (J.Y.J.)
[\*] Correspondence: jylee7@skku.edu (J.Y.L.); withs@kitech.re.kr (C.K.L.)

**Abstract:** Products are manufactured as printed electronics through electro-conductive ink having properties suitable for flexible substrates. As printing process conditions affect the quality of the electronic properties of the final devices, it is essential to understand how the parameters of each process affect print quality. Spray printing, one of several printing processes, suits flexible large-area substrates and continuous processes with a uniform layer for electro-conductive aqueous ink. This study adopted the spray printing process for cellulose nanofiber (CNF)/carbon nanotube (CNT) composite conductive printing. Five spray parameters (nozzle diameter, spray speed, amount of sprayed ink, distance of nozzle to substrate, and nozzle pressure) were chosen to investigate the effects between process parameters and electrical properties relating to the morphology of the printing products. This study observed the controlling morphology through parameter adjustment and confirmed how it affects the final electrical conductivity. It means that the quality of the electronic properties can be modified by adjusting several spray process parameters.

**Keywords:** spray parameter; surface morphology; electrically conductive print





## 1. Introduction

With the recent development of smart devices, electronic products require light and flexible properties [1,2]. To meet this demand, the printing system that uses electrically conductive ink on a thin film over a large area is used [3]. It has potential application including light-emitting diodes [4], thin film transistors [5], flexible and conformal antenna arrays [6], radio-frequency identification [7], photovoltaic devices [8], electronic circuits fabricated in clothing [9], and biomedical devices [10]. Various conductive materials are used for conductive ink, such as conducting polymer [11], metal [12], and carbon nanomaterials [13]. Among them, carbon nanomaterials can be a good choice as the active materials of conducting ink. Especially, carbon nanotubes (CNTs) possess high flexibility [14], low mass, and a large aspect ratio (typically >1000) [15]. Additionally, it can transport electrons over long lengths without signature interruption [14]. Especially, its fibrous feature helps to obtain an excellent electrical conductivity additionally. Meanwhile, cellulose nanofibers (CNFs) have attracted considerable attention as the most potent potential feedstock for bio-based polymeric material production [16]. In particular, cellulose nanofibers may stand out due to their application to polymer matrix with mechanical strength, porosity, high moisture retention, high surface functionality, and tangled fiber networks [17]. They are used as a fibrous element in electric devices such as foldable displays, implantable medical devices, touch panels, wearable energy storage devices, and various sensing devices [1]. In the electrically conductive aqueous ink composed of CNF and CNT selected for this study, a composite of CNF and CNT with a diameter of several to tens of nanometers is dispersed in water [18]. The dispersed conductive materials, a composite of CNF and CNT, are linear in fiber form. It can help electrons move through electricity, improve electrical conductivity, and cope with various shape deformations such as bending, stretching, and

twisting [19,20]. In general, the larger aspect ratio of fibrous conductive materials causes the higher conductivity of coated thin films [21]. However, their fibrous structure causes higher surface tension [22] and hairy form in the solvent, so it is important to decide on suitable coating methods and process parameters for good electrical properties. The properties and quality of the final coating are strongly influenced by the coating method and process conditions [23].

Printing is a kind of solution-based deposition process such as spin coating, spray, dip coating, and inkjet methods. For printed electronics, inkjet printing is the most suitable method of sending ink through a long nozzle and has limitations for printing ink composed of fibrous structured materials [24]. Spray coating is usually used for printing on a large area that is less affected by the structural characteristics of ink by quickly sending ink under pressure. Spray printing is a viable, scalable technique for the large-scale, fast, and low-cost fabrication of solution processing. It has been widely used for device fabrication, despite the fundamental understanding of the underlying and controlling parameters [3].

In the spraying process, the printing ink undergoes three steps of atomization, deposition, and coalescence from the spray nozzle to the substrate as shown in Figure 1 [25] Atomization is the process of transforming a liquid into a spray of fine particles in the surrounding gas, that is air. The breakdown of the liquid into small particles is achieved when air mixes with the liquid. A spray nozzle is used to generate the atomized spray, which passes through an orifice at high pressure and in a controlled manner. This process is widely utilized when distributing material over a cross-section area or generating a liquid surface area over an object. Obtaining fine droplets depends on the nozzle shape, the physicochemical properties of the droplets (surface tension, viscosity, density), air-to-ink mass ratio, and spraying air pressure. Second, deposition is the process of droplet impingement on the surface. The impingement of the atomized droplets without rebounding or splashing is favorable in the interest of obtaining better printing intensity and stability. The deposition is affected by the spray parameters such as spray velocity, spray density, and distance of nozzle-to-surface (DNS). While high spray speeds have been reported to facilitate the spreading of droplets upon impingement on the surface, too high a velocity can result in droplets being blown off the surface by the stream of atomizing air. Finally, after deposition, the coalescence between droplets follows. The shape of coalescence is further affected by the substrate's composition and the factors that influence the above steps.

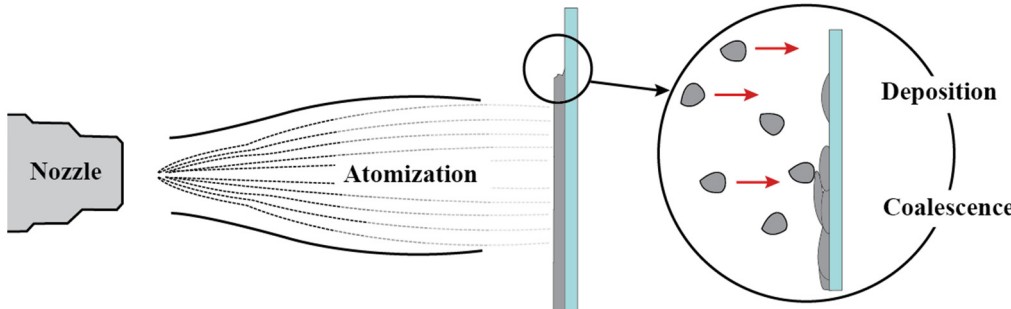

**Figure 1.** Three steps of the spray process.

The printing surface sprayed over the three steps has particular morphological characteristics such as surface roughness and shape [25]. As the thin film's surface roughness increases, the resistivity increases, decreasing its electrical conductivity [19]. The surface roughness is combined with the intrinsic electrical properties of the sprayed conductive material and ultimately affects the electrical properties of the conductive printing layer [20,26]. Therefore, for the high electrical conductivity of the final print, it is essential to control the surface roughness through conditional control in the spray printing process.

This study fabricated a flexible and electrically conductive print using a CNT/CNF aqueous ink spray printing system. In this process, we investigated the effect of various spray parameters on the surface roughness and the electric conductivity of final prints.

During the spray process, the parameters are organized into five types: nozzle diameter, spray speed, amount of spray ink, the distance of the nozzle to substrate, and spray pressure. Each parameter was controlled independently during the substrate's CNT/CNF ink printing.

The effect of each spray parameter condition on the print surface morphology was studied by mainly confirming a change in surface roughness of the final print surface. In addition, the change in electrical conductivity of the print was also exhibited. Thus, the correlation between the morphological characteristics and electrical conductivity was confirmed.

## 2. Materials and Methods

### 2.1. Materials

CNTs were purchased from OCSiAl (TUBALL, carbon >75%). The eucalyptus hardwood bleached kraft pulp (NIST® RM 8496, Sigma-Aldrich Co., Ltd., St. Louis, MO, USA) was purchased for manufacturing CNF. In total, 100 μm of the thickness of a polyethylene terephthalate (PET) film (SG05, SKC, Seoul, Korea) with corona treatment was used as a substrate.

### 2.2. Fabrication of Electro-Conductive Ink

The eucalyptus hardwood bleached kraft pulp and SWCNTs were added to DI water. They were subsequently ground in a mechanical milling system at 10 °C for 2 h. The proportion of CNF and CNT in conductive ink was 10:1. The viscosity was measured using a Brookfield viscometer (LVDV-II) and the spindle LV-1. The viscosity of the ink is 1762 cP.

### 2.3. Spray Printing

We used a spray coater with an air dispenser type (multi spray coater, Hantech, Daejeon, Korea) for the spraying process with a hollow cone nozzle. This machine is a customized model with a moving substrate added. The structure is presented in Figure 2. The machine can adjust the spray to make samples. The machine sprays the liquid using air pressure by adjusting 5 parameters such as nozzle pressure, moving speed, and several spray counts. During the spraying process, the spray nozzle is fixed at an angle of 10 degrees. After spraying, the samples were allowed to dry at room temperature for 24 h before analysis. The polyethylene terephthalate (PET) film of 10 × 10 cm was used as a substrate. Finally, spray-print samples were obtained according to a parameter change covering the entire surface of the PET substrate.

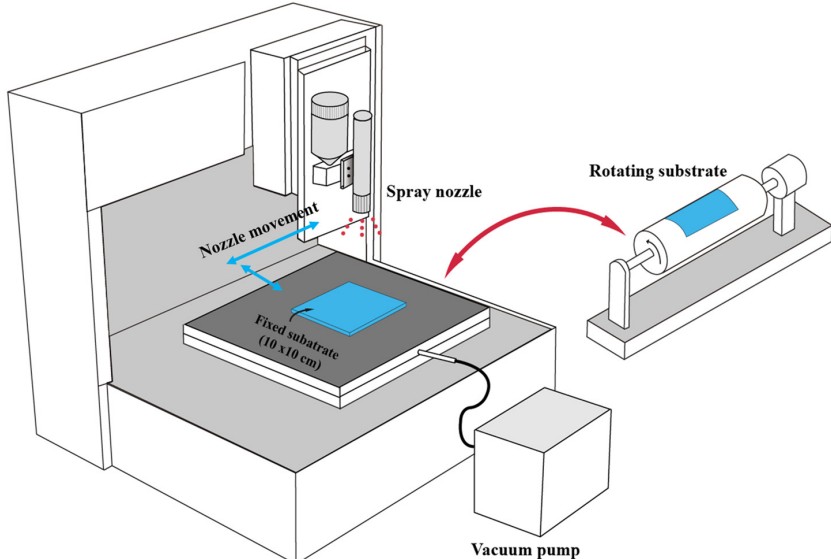

**Figure 2.** Spray coater machine with an air pulse dispenser type.

*2.4. Characterization*

The print's surface roughness was visually assessed by a 3D laser scanning microscope (Keyence, VK-X200K, Seongnam-si, Korea) with and magnification of $10\times$ using a violet laser at 408 nm. Thickness was determined by using a micro-thickness gauge VL50A from Mitutoyo. The overall area thickness of the print was determined by dividing the substrate film of $10 \times 10$ cm size into 20 points at equal intervals. FE-SEM (Hitachi, SU8020, Tokyo, Japan) was used to obtain the print's surface microstructure. The surface resistance (ohms/sq) and conductivity (S/cm) were measured and calculated using a surface resistance analyzer (AiT, CMT-SR2000N, Suwon-si, Korea) using a 4-point probe method for measuring electrical characteristics. The conductivity (S/cm) was expressed only by inverting the resistivity (ohms·cm). The surface resistance analyzer measured the resistivity using the thickness of the spray-print layer, which means that the conductivity is affected by the thickness, unlike the surface resistance. The bulk resistance was measured directly by digital multi-meter (Keysight, 34461A, Santa Rosa, CA, USA). The shape of deposited droplets and coalescence was defined using Image J software (Leica, Nusloch, Germany).

## 3. Results

We investigated the effect of the various spray-processing parameters on print morphology and electrical conductivity. There are five adjustable parameters. These parameters are independent variables that can control morphological factors such as roughness and high uniformity of spray-printed surfaces. Surface roughness is expressed in Ra values using a 3D laser scanning microscope. Ra is a method of measuring the average roughness value relative to the centerline. The sample was adjusted for only one factor to be compared among the five parameters, and the remaining four variables were applied in the same way. Table 1 shows the parameters and values selected.

**Table 1.** Five parameters of the spray process.

| Spray Parameter | Nozzle Diameter (mm (Ø)) | Spray Speed (mm/s) | The Amount of Sprayed Ink (mL) | Distance of Nozzle to Substrate (cm) | Nozzle Pressure (kPa) |
|---|---|---|---|---|---|
| Value | 0.4, 0.7 | 180, 190, 200, 210, 220, 230, 240, 250, 500, 1000, 2000 | 5, 10, 15, 20, 25 | 10, 12, 14, 16, 18 | 50, 55, 60 |

*3.1. Effect of Nozzle Diameter*

The nozzle diameter is the first factor in determining the effect of the spray process on the morphology of the printing surface. Nozzle size influences the formation of particle size critically. The particle size affects how the CNT/CNF complex in the fiber form of conductive ink is distributed on the substrate. Various spray nozzle designs are used, such as full cone, hollow cone, and flat fan. A previous study reported that droplets are commonly the largest size in a full cone and the smallest in a hollow cone [23]. In this study, a hollow cone nozzle was used for the experiment. Two nozzle diameters, 0.4 Ø and 0.7 Ø, compatible with spray machines, were used to make samples at a distance of 14 cm for one second. Figure 3 shows the deposition and coalescence of droplets according to nozzle diameter, and the number of droplets was calculated by simplifying the image. In Figure 3a,c, the dark areas are ink, and the light areas are substrate. Figure 3b,d are the same as Figure 3a,c, respectively, but also show only the outline of deposited ink, which was converted by Image J. The criterion for selecting 'deposition' is not size but whether or not the droplets overlap; therefore, Figure 3b,d show a more apparent difference between the deposition and the coalescence.

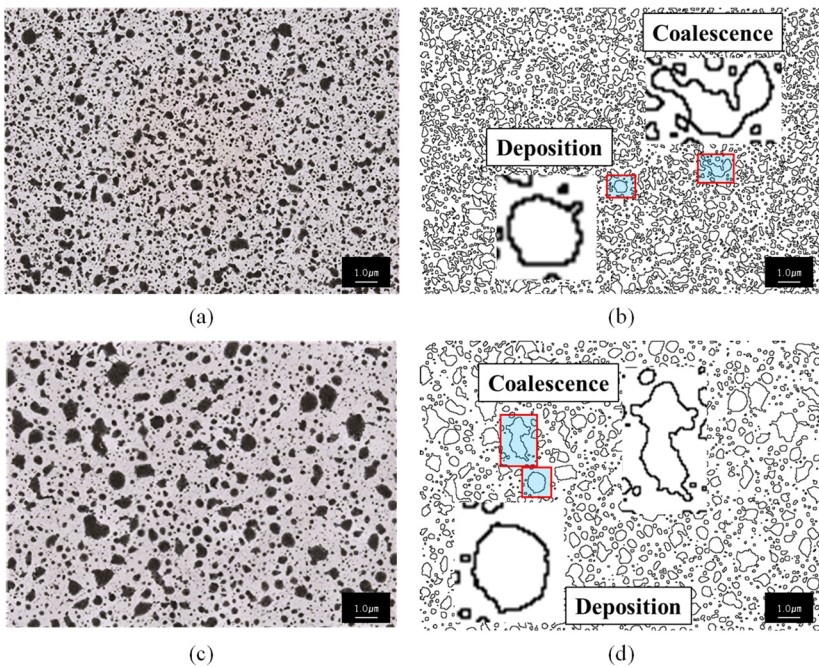

**Figure 3.** Effect of nozzle diameter on atomized ink droplet figures and clustering shape (deposition and coalescence). The 0.4 Ø nozzle is shown in (**a**,**b**), the 0.7 Ø nozzle in (**c**,**d**).

In Figure 3, the number of droplets in the 10 × 10 cm film was 1755 and 1252 for the samples of 0.4 Ø and 0.7 Ø, respectively. The 0.4 Ø sample has 40.18% more drops than the 0.7 Ø sample. In addition, it was confirmed that the sum of the droplet area was 62,092.52 $\mu m^2$, which was 212.06% larger than the 0.7 Ø nozzle of 19,897.40 $\mu m^2$. The results can be seen as the nozzle diameter that controls the sprayed droplet's size determines the size, pattern, and distribution of the atomized droplets on the substrate [23]. In addition, the images in Figure 3b,d were analyzed to determine the morphological characteristics of deposition and coalescence of atomized droplets. Droplet deposition and coalescence shape showed differences in size. It confirms that nozzle diameter does not affect deposition and coalescence during the spray process but only affects atomization.

In Table 2, the surface roughness of 0.4 Ø is 0.95 μm and 0.7 Ø is 1.03 μm. Therefore, 0.4 Ø is less than 0.7 Ø, and the morphological characteristic is more uniform at 0.4 Ø. According to this morphological characteristic, the spacious uniform printing of CNT/CNF composite ink with a tiny droplet is expected to benefit the electrical characteristic because the improved connection between the droplets leads to an increase in the conductivity of printing [27].

**Table 2.** Number of holes and surface roughness result by nozzle diameter.

|  | 0.4 Ø | 0.7 Ø |
|---|---|---|
| Number of droplets | 1755 | 1252 |
| Surface roughness (μm) | 0.95 | 1.03 |

### 3.2. Effect of Spraying Speed

The effect of the spray speed on the resulted printing was confirmed. The nozzles of Figure 4 showed that it is advantageous to spray droplets evenly and widely onto the substrate in a tiny size for uniform printing. It is predicted that printing speed also would have an impact on morphology. The printing speed means nozzle movement speed. There are two methods of spray used: one is the nozzle moving, and the other is the substrate moving, as shown in Figure 4. The printing speed for the nozzle moving was carried out at intervals of 10 mm/s from 180 to 250 mm/s, which is an operable range from the spray

machine. The substrate moving was used for high-speed printing with a fixed spray nozzle, and the substrate attached to the roll was allowed to rotate up to 2000 mm/s.

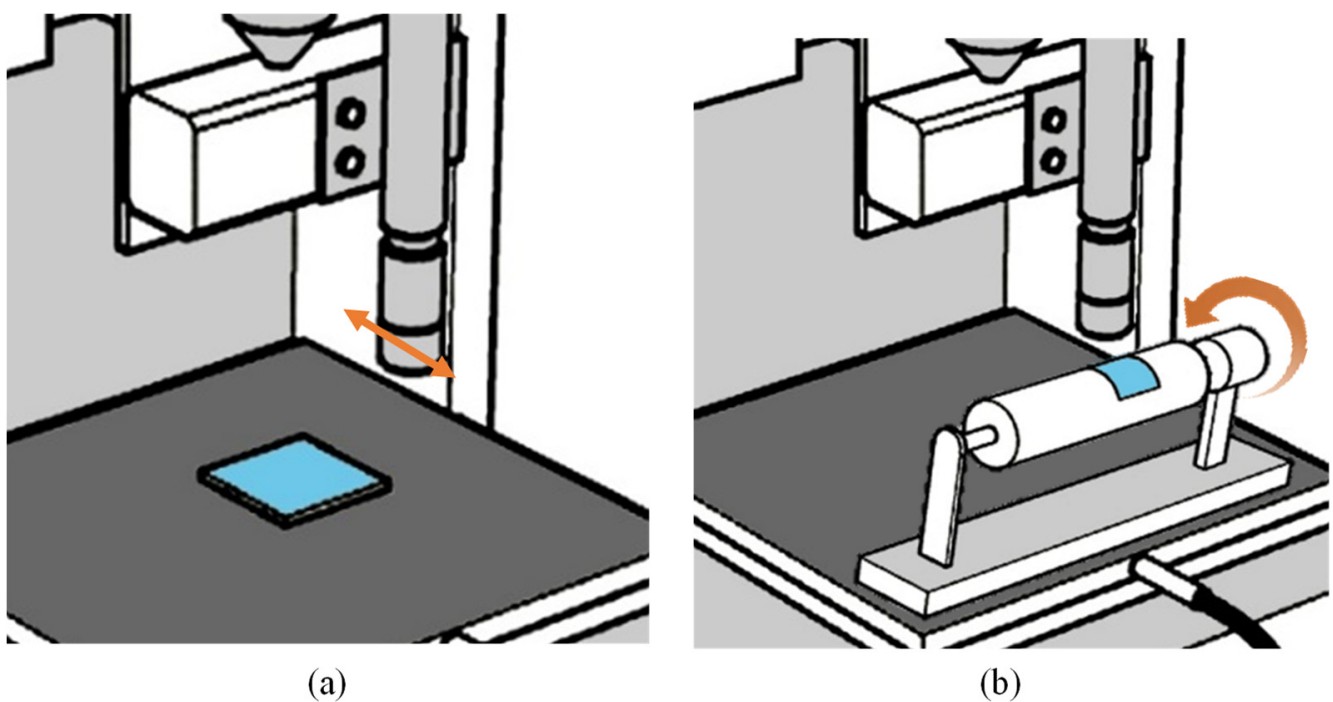

**Figure 4.** Spray procedure by using a multi-spray coater ((**a**): low speed, (**b**): high speed).

The printing was carried out on a substrate moving from 200 to 2000 mm/s. In both cases, the distance between the nozzle and the substrate was 14 cm. Finally, a sample was prepared by fixing the ink injection amount to 20 mL.

Finally, a sample was prepared by fixing the ink injection amount to 20 mL, and 5 mL was printed a total of 4 times.

Figure 5 shows the effect of spray-printing speed, although different printing methods were used at low speeds (below 250 mm/s) and high speeds (over 500 mm/s). In the case of low-speed printing, the nozzle moves, and in the case of high-speed printing, the drum rotates. Figure 5a shows that the surface roughness decreased by 40% from 1.1 to 0.57 μm, increasing speed from 180 to 250 mm/s. However, it was found that a change (less than 1%) in surface roughness at speeds of 500 mm/s or more was negligible. The decrease in surface roughness due to increased print speed under fixed-parameter conditions can be explained. First, the amount of ink sprayed on the substrate per unit time is small. In this case, since the droplets on the film substrate are quickly dried, the change in the deposition morphology is small. There is a result of reducing the effect on coalescence after deposition [25]. Second, the morphology of the deposited droplets has directionality by the direction of movement of the nozzle or rotation of the substrate. In particular, the effect is expected to be remarkable for fiber-based ink. When the electro-conductive nanofibers in droplets are arranged, defects such as a coffee ring structure that increases surface roughness are reduced and uniform [28,29]. However, as shown in Figure 4, it is found that the print surface morphologies are not affected at a certain speed or higher.

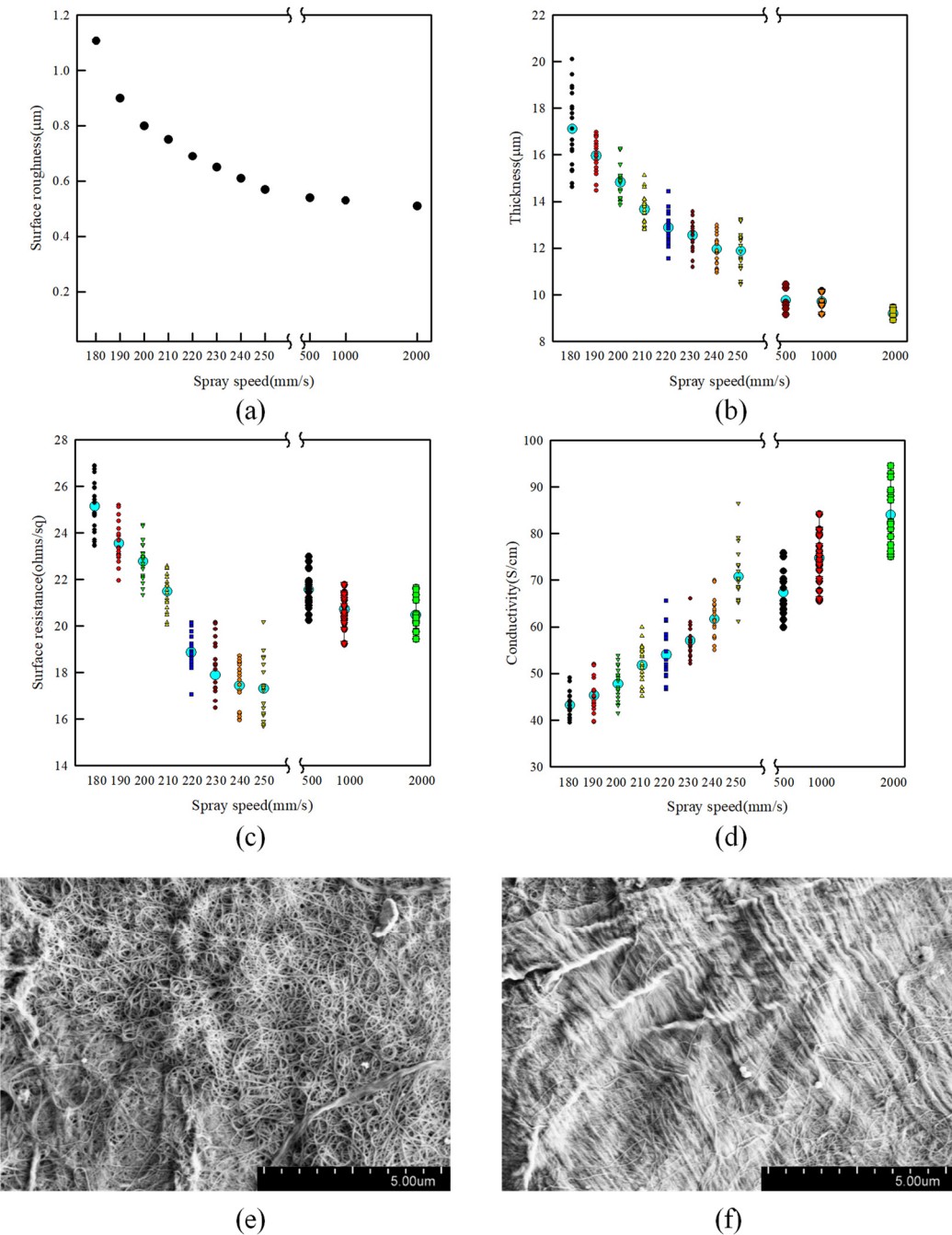

**Figure 5.** Total properties by controlling spray speed (**a**) surface roughness, (**b**) thickness, (**c**) surface resistance, (**d**) conductivity, and scanning electron microscope (SEM) images of sprayed film at speeds of (**e**) 200 mm/s and (**f**) 2000 mm/s by controlling spray speed.

The average thickness was measured at 20 points in the sample at equal intervals of 2 cm, and the thickness distribution confirmed the uniformity of the spray-printed layer. As shown in Figure 5b, as the printing speed increased, the thickness significantly decreased from 180 to 220 mm/s, and there was a threshold of 240 mm/s. In addition, even at high speed of 500 mm/s or more, there was no significant thickness change, and the deviation decreased. As shown in Figures 4f and 5e, it can be understood that the electro-conductive nanofibers in the droplets adhere to each other as the printing speed increases, and thus the thickness decreases.

Figure 5c,d show a change in electrical characteristics according to a printing speed. The surface resistance in Figure 5c shows a distinct reduction of about 31.17% from 25.15 ohms/sq in 180 mm/s to 17.31 ohms/sq in 250 mm/s. On the other hand, it was found that there was little change in surface roughness at a speed of 500 mm/s or more. As shown in Figure 5d, the electrical conductivity was increased by 63.71% up to 250 mm/s, which is the same behavior as the change in surface resistance. However, unlike the surface resistance, the electrical conductivity continuously increased to 24.68% at a printing speed of 500 mm/s or higher. This result was understood to be directly affected by the thickness of the printing layer through the thickness change behavior in Figure 5b. Normally, as the number of conductive materials increases, the electrical conductivity increases. However, the increase in electrical conductivity despite the thinning of the printing layer in spray printing can be explained as a complex effect of decreasing the thickness and surface resistance as the printing speed increases. In addition, the SEM images of Figure 5e,f show that the fibrous conductive material is arranged in a rotation direction of 2000 mm/s. The printing speed can form the fibrous conductive material's direction. Based on this, the conductive material constituting the printing layer is advantageous for improving electrical conductivity when it has directionality [30].

### 3.3. Effect of Spray Amount

In spray printing, the amount of spray is an essential factor in determining the thickness of the printing layer. Since the thickness of the printing layer has a reliable effect on electrical conductivity, electrical characteristics according to the amount of spray are confirmed through surface roughness and thickness changes. Therefore, the spray amount was set at intervals of 5–25 mL using the nozzle 0.4 Ø, DNS 14 cm, pressure 50 kPa, and printing speed as fixed.

As the spray increased in Figure 5a, the surface roughness decreased from 1.28 to 0.85 μm, the most significant decrease at 10 mL. An increase in spray leads to printing several times in the same area. Repetitive spray printing means the superposition of the coalescence layer by the deposited droplets. It was confirmed that as the coalescence layer accumulated, the even surface of the printing layer was obtained. For the surface roughness, the increase in the amount of spray showed the same result as the increase in the printing speed; the lower the surface roughness, the lower the surface resistance. As shown in Figure 5c, the surface resistance decreases 90.4% from 70.24 ohms/sq of 5 mL to 6.75 ohms/sq of 25 mL. The surface resistance was decreased proportionally to the spray amount. When only the spray amount and the printing speed can be controlled, it was found that it is advantageous to obtain better electrical conductivity by adjusting the spray amount.

The thickness change in Figure 6b represents a threshold of 20 mL, and the thickness of 25 mL is almost the same as that of 20 mL. As shown in Figure 6a,b, it can be seen that thresholds of surface roughness and thickness are different. This behavior is the same as the electrical behavior shown in Figure 6c,d. The threshold of surface roughness and surface resistance is 10 mL, and the threshold value of thickness and electrical conductivity is 20 mL. This result shows that it is more effective to adjust the surface roughness to decrease the surface resistance when spray printing. Likewise, the thickness is more effective in increasing the electrical conductivity. The conductivity increased 69.73% from 31.12 S/cm of 5 mL to 52.82 S/cm of 25 mL in Figure 6d. The results also showed that the conductivity was increased proportionally to the spray amount.

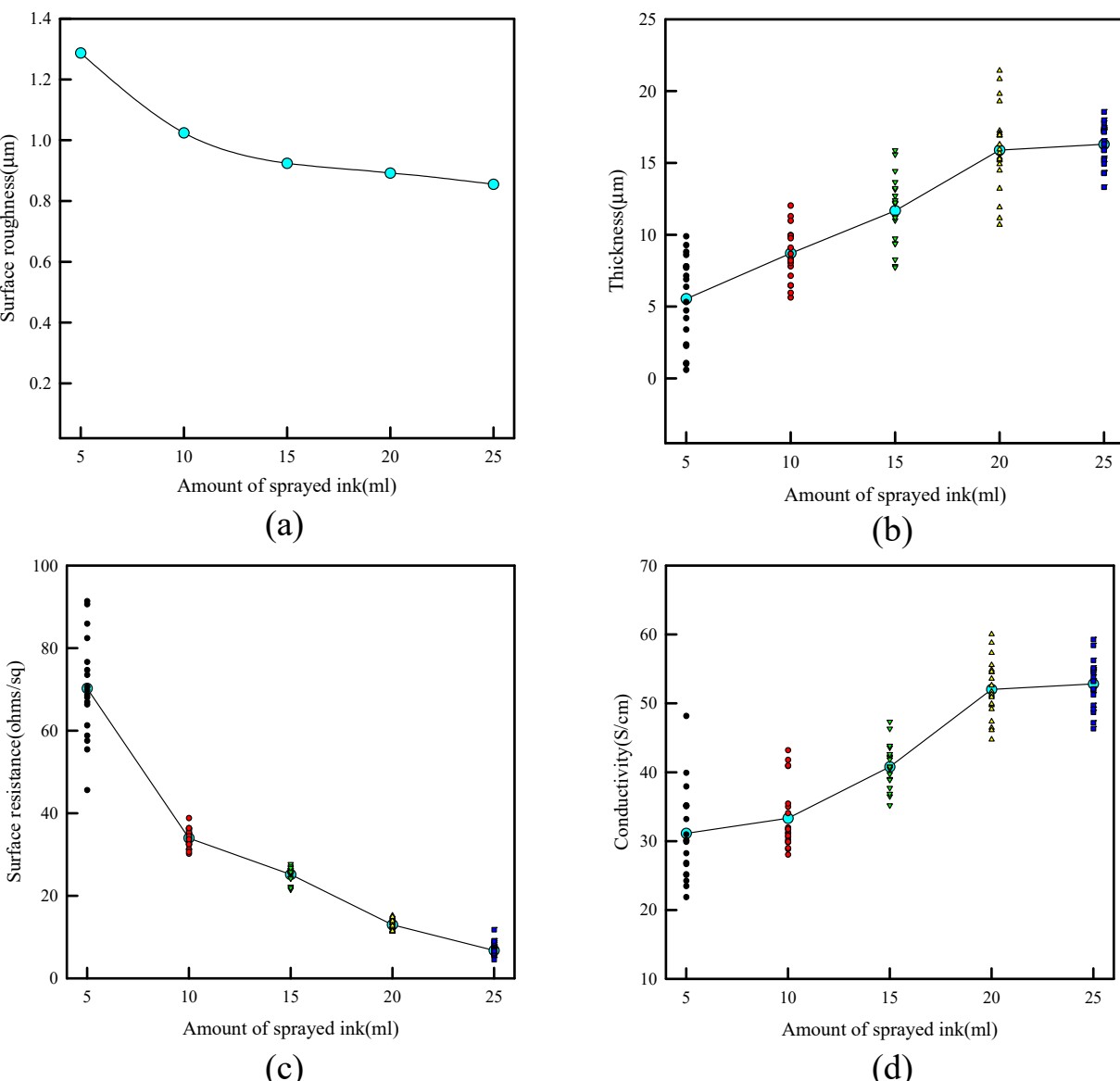

**Figure 6.** Total properties according to amount of sprayed ink: (**a**) surface roughness, (**b**) thickness, (**c**) surface resistance, and (**d**) conductivity.

*3.4. Effect of Distance of the Nozzle to Substrate (DNS)*

The effect of the distance DNS from the nozzle to the substrate on the printing surface roughness and electrical characteristics was confirmed.

DNS determines the deposition location of ink droplets because droplets discharged from the nozzle fly to the printing surface, and depending on the shape of the nozzle used, specific shapes may be created on the printing surface, such as coffee rings. A sample was prepared by spraying within 10 s using a 0.4 Ø hollow conical nozzle to study the shape of the spray ring by DNS. The shape of the formed spray ring is expressed as a height profile of the cross part.

DNS used were 10, 12, 14, 16, and 18 cm, with the surface roughness of 2.09, 1.45, 1.34, 1.25, and 1.04 mm, respectively. Figure 7a shows that the surface roughness decreases as DNS increases. As DNS increases, the final velocity of droplets sprayed from the nozzle decreases. The reduced velocity can explain the decrease in surface roughness by inducing the impact of atomized drops without recoil or splashing. In Figure 7b, as DNS increased to 14 cm, the bulk resistance decreased but increased again from 16 to 18 cm. When the

DNS is excessively increased, water droplets are lost out of the substrate, and as a result, the bulk resistance gradually increases.

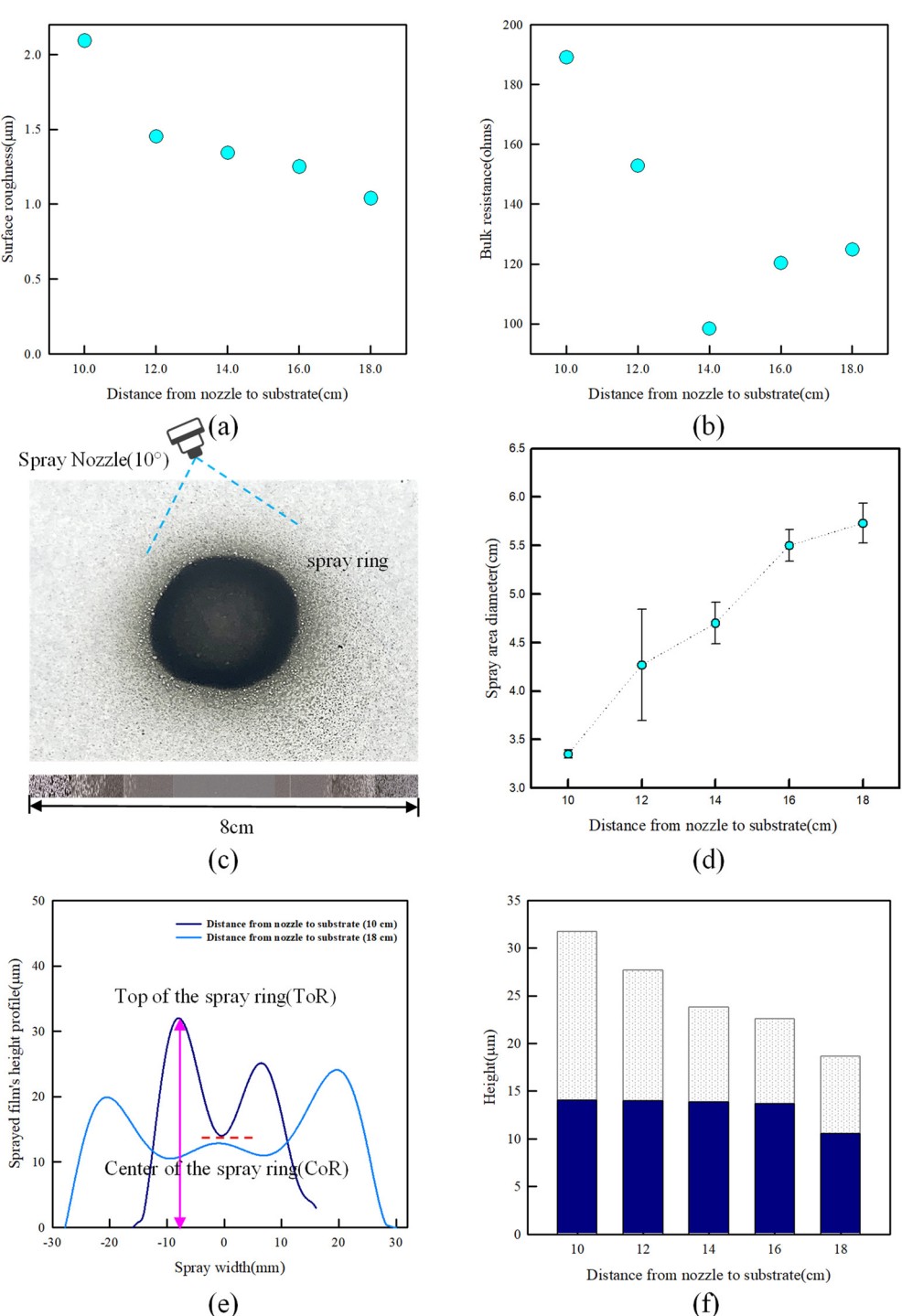

**Figure 7.** Effect of distance of nozzle to substrate in (**a**) surface roughness, (**b**) bulk resistance, (**c**) spray ring image in DNS 10 cm, (**d**) spray ring diameter, (**e**) height profile of spray ring, and (**f**) a comparison of ToR and CoR.

Figure 7d shows that as DNS increases, the diameter of the spray ring also increases. The diameter of the spray ring increased linearly to 76.47%. In Figure 7e, the height profile of the spray print was measured using a 3D laser microscope. The donut-shaped cross-section in the height profile was obtained, where the highest part (ToR) is ring-shaped, and

the lowest part (CoR) is centered. Figure 7e,f show that the maximum height (ToR) of the spray ring decreases as DNS increases. Increasing the dispensing distance when spraying the same amount of ink means that the printing area is enlarged, and the surface is evenly printed. The decrease in ToR also tended to decrease linearly and decreased by up to 15%.

On the other hand, it is shown in Figure 7f that the ToR change in DNS of 18 cm in DNS of 10 cm is 40.43%, but the CoR change of the spray ring is only 2.46%. Interestingly, the central height of the spray ring, CoR, was almost constant independently of DNS. These results mean that a uniform surface can be obtained using CoR when manufacturing a large-area print.

### 3.5. Effect of Nozzle Pressure

Pressure on the nozzle is an essential factor in spray printing that determines the velocity and area of water droplets.

Figure 8 shows the height profile of the spray ring according to spraying pressure, surface roughness, and electric resistance. The pressures used were 50, 55, and 60 kPa, and the surface roughness was 1.1531, 1.3443, and 1.4013, respectively. As shown in Figure 8a,b, as the spraying pressure increases, the surface roughness increases, and accordingly, the resistance of print increases. Surface roughness increased by up to 21.5%, and electric resistance increased by 44.5%. Figure 8c,d show the effect of spray pressure on electrical characteristics more clearly with spray ring height profiles. As the spraying pressure increased, the difference between ToR and CoR increased, and the height difference of printed surfaces at a spraying pressure of 60 kPa was 10 μm or more. It explains the cause of the significant change in resistance compared to surface roughness. It was confirmed that the change in diameter of the spray ring was more directly affected by DNS than by spray pressure.

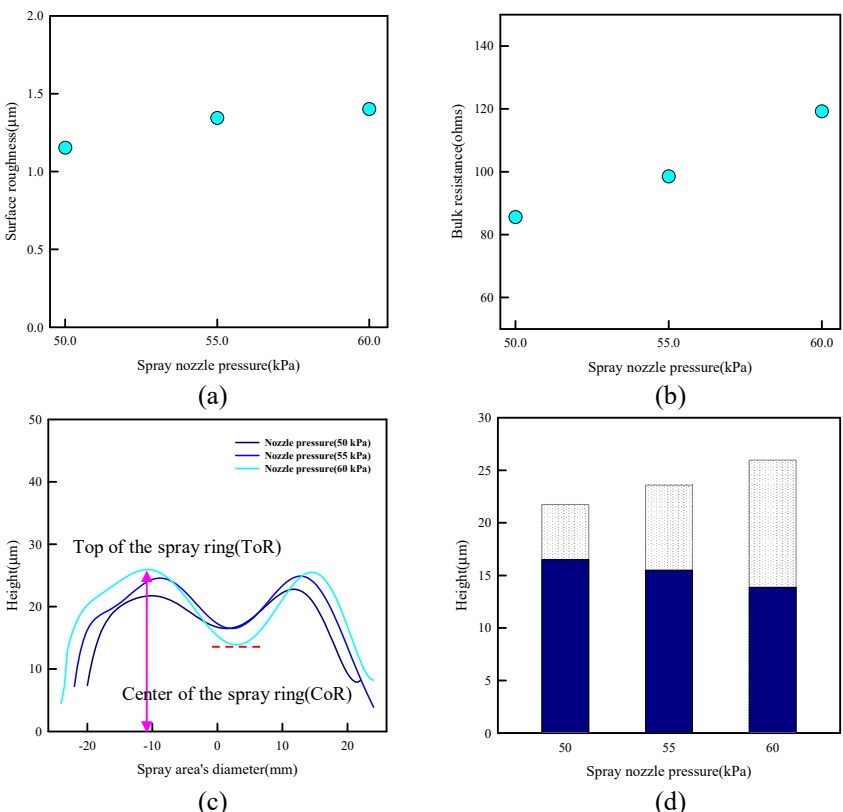

**Figure 8.** Effect of spray nozzle pressure in (**a**) surface roughness, (**b**) bulk resistance, (**c**) height profile of spray-ring, and (**d**) a comparison of ToR and CoR.

## 4. Conclusions

The shape, surface roughness, and thickness of spray printing were studied by changing five parameters applied to the spray printing process. The results revealed how the five parameters affect the electrical properties of the spray printed. It was confirmed that the smaller the nozzle diameter, the smaller the size of the deposited droplet, and the printing surface was even. Furthermore, as the printing speed increases, the thickness and surface roughness of the printing layer decrease, surface resistance decreases, and electrical conductivity is improved. In the case of spray amount, each role of the electrical characteristics was confirmed by checking the behavior of the thickness and surface roughness of the spray printing layer. In addition, characteristics of the spray printing pattern shape were found through the study on the spray distance and pressure. Finally, the method of improving electrical characteristics was suggested by adjusting the five parameters for obtaining a uniform printing surface.

**Author Contributions:** Author Contributions: Conceptualization, C.K.L. and G.H.K.; methodology, C.K.L.; validation, G.H.K., E.A.S., J.Y.J. and J.Y.L.; formal analysis, G.H.K. and E.A.S.; investigation, J.Y.J.; resources, C.K.L.; data curation, G.H.K., E.A.S. and J.Y.J.; writing—original draft preparation, G.H.K.; writing—review and editing, J.Y.L. and C.K.L.; visualization, G.H.K.; supervision, C.K.L. and J.Y.L.; project administration, J.Y.L. and C.K.L.; funding acquisition, C.K.L. All authors have read and agreed to the published version of the manuscript.

**Funding:** This study was conducted with the support of the Korea Institute of Industrial Technology as "Development of microfactory-based technology for future smartwear manufacturing (kitech EH-22-0003)".

**Institutional Review Board Statement:** Not applicable.

**Informed Consent Statement:** Not applicable.

**Data Availability Statement:** Not applicable.

**Conflicts of Interest:** The authors declare no conflict of interest.

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
