# Peer review of "Effect of Spray Parameters on Electrical Characteristics of Printed Layer by Morphological Study"

_processes, doi:10.3390/pr10050999_

Round 1

Reviewer 1 Report

Your article addresses the question of spray parameters, picking five of them for your investigation. The topic is actual: generating a conductive layer on flexible substrates. However, many applications require a patterned conductive layer. With spray coating only homogeneous layers can be generated. I suggest to attract more readers with defining a use case additional to your investigations.

There are some remarks, suggestions and hints for your paper. "L" is indicating the line number.

General:

  • There has to be a space between value and unit, e.g. 1 h (not 1h), 5 cm (not 5cm), ... In most of the cases this space is missing
  • In Chapter 2 I miss the information about the resulting sample. Is it a line of material? What is about surplus material not hitting the sample but the nearby area? What does the setup look like? Maybe you can refer to Figure 3 if appropriate?

Specific:

L39:  There is a misspelling "[1] ]". Please correct.

L44: "large area deformation" is unclear to me. I think you mean something like bending, stretching, ... Maybe you can choose another wording?

L59: Figure 1 is given without any special explanation. You should refer to it in the text.

L63: you define the medium for generating spray of a liquid as "gas" - but in L64, L68, L69, and L75 you use "air" instead. You may add one sentence for interconnecting "gas" and "air".

L105: If you mix 1 wt% pulp and 0.1 wt% CNT, for me it is something like 10:1. In L105 you give "9:1". Is there any explanation?

L109: The device of air pulse dispenser is not given, only the company and the country

L115: The PET film is 10 cm x 10 cm. This is a mismatch with e.g. Figure 6 (d), where you have 12 cm diameter. Please adjust or add a special case.

L124: All variables are well explained, but I miss the meaning of "S" and "pl".

L137: "table1" should look like "Table 1"?

L138: There is a "1" surplus.

L138: What dimension has the nozzle diameter? Is it µm, mm, cm, ...?

L138: If I look for the parameters, at least a variation of a factor of 1.8 is done - except for the nozzle pressure. Is there any technical reason why the variation is only a factor of 0.2?

L151: I miss a scale bar to decide about the sizes. I assume that dark areas are ink and light areas are substrate? You may indicate in your text. Additionally, there are many smaller dots in (b) and (d) than that you have selected as "deposition". Please give an explanation for the generation of these. The other option would be to have an analysis of single spray dots in another experimental setup.

L155: You compare number of drops. Unless you explain smaller droplets in Figure 2, I'm not sure about your assumption.

L188: You claim a ink amount of 20 mL. Please explain about distribution: What amount is deposited on the substrate? What amount get lost? How many times the sample is printed to deposit this amount of ink? What is about intermediate drying? What is about time required for printing?

L190: The image quality needs improvement. This remark might become obsolete if the quality is decreased for the review process. It the same with other graphs.

L190: The x-axis is unclear. It seems to be linear but with different scaling besides the break?

L190: There seems to happen a totally different behavior in dependence of the Figure 3 experimental setup. Otherwise there would be no break in values in (b)-(d). Is there any additional drying process happening?

L196: "," should be placed the line before

L208/209: Please review this sentence. It seems to me that the first "20" is wrongly given?

L218: Personal feeling: I would feel more comfortable if the result like 25 Ohms/sq is given before the input value of 180 mm/s rather than vice versa.

L241: If the formula you give in L124 is correct, then L = 1/R. This would mean that there is a mismatch in your graphs (b), (c), and (d). If the sheet resistance in (c) decreases by a factor of ~ 2 than the conductivity need to increase by a factor of ~2. But it stays constant. Rather it seems that the conductivity follows the behavior of the thickness (b). This doesn't give any sense to me. Could you please explain why sheet resistance and conductivity show such behavior? Or did you do not use the formula L124 for your results?

L255: Your conclusion "it can be seen that the effect of the spray amount is greater than that of the print speed". In my opinion this needs to be referred to the experimental conditions. It is a question of the parameters you apply. Therefore, please rethink your phrasing.

L277: (c) a scale bar is missing

L277: (b) is "resistance" the right word. Before you have given "sheet resistance". I do not find any explanation of your measurement setup for "resistance"

L277: in (a) and (b) you give e.g. 18.0 while in (d) and (f) only 18. Please consolidate

L277: I don't know what "ToR" and "CoR" mean and I cannot find any definition.

L284: You claim that there is a decrease of velocity of droplets when DNS increases. Is it your assumption, or is there any measurement or theory?

L287: What does "bulk resistance" mean? It is only used in L287 and L289 - but without any explanation, definition. How is it measured?

L306: (b) as L277

Author Response

첨부파일을 참조하시기 바랍니다.

Reviewer 2 Report

Article "Effect of Spray Parameters on Electrical Characteristics of Printed Layer by Morphological Study" is interested specially for application involved roll to roll printing. The material system used is also interesting however the article lack the following:

1. Potential application

2. Effect of substrate surface

3. Binding energy of the sprayed material

4. Device design used for conductivity measurement

5. Selection of solvent and material systems reason of choosing an A speed vs B.

6. Evaluation if the proposed conclusion is only valid for a material system or can be extended to other compositions

7. introduction section need a deeper review specially to include the application

"Carbon nanotubes, orange dye, and graphene powder based multifunctional temperature, pressure, and displacement sensors June 2020Journal of Materials Science: Materials in Electronics 31(4) (DOI: 10.1007/s10854-020-03424-5) is one such example.

Addressing the above will add value to this work.

Author Response

첨부파일을 참조하시기 바랍니다.

Round 2

Reviewer 1 Report

Dear authors,

thank you for following up with my comments. From my feeling, you have improved the quality of your paper.

You have raised the question about Figure 5. My recommendation is that you keep the splitted x-axis rather than a linear one. You have added a sentence to your text explaining the split; so it is fine.

There is a problem arising for displaying the character Ω in your revised paper:

  • y-axis of Fig. 5 (c)
  • y-axis of Fig. 6 (c)
  • y-axis of Fig. 7 (b)
  • y-axis of Fig. 8 (b)

All the best for further research on this topic.

Author Response

Thank you for your kind comment. 
According to your comment, we checked and corrected character Ω to ohms in figure 5(c),6(c),7(b) and 8(b). Also, the manuscript of line 159, 162 on page 4 and line 308, 309 on page 10 was corrected character Ω to ohms.

Reviewer 2 Report

Accepted 

Author Response

Thank you for your comments. As your comments, we improved the quality of paper.